# Severe Asthma Remissions Induced by Biologics Targeting IL5/IL5r: Results from a Multicenter Real-Life Study

**DOI:** 10.3390/ijms24032455

**Published:** 2023-01-27

**Authors:** Angelantonio Maglio, Carolina Vitale, Corrado Pelaia, Maria D’Amato, Luigi Ciampo, Eliana Sferra, Antonio Molino, Giulia Pelaia, Alessandro Vatrella

**Affiliations:** 1Department of Medicine, Surgery and Dentistry “Scuola Medica Salernitana”, University of Salerno, 84100 Salerno, Italy; 2Department of Health Sciences, University “Magna Græcia” of Catanzaro, 88100 Catanzaro, Italy; 3Department of Respiratory Medicine, Federico II University, 80100 Naples, Italy

**Keywords:** severe asthma, OCS-dependent asthma, asthma remission, mepolizumab, benralizumab, airway inflammation, anti-IL 5

## Abstract

Add-on biological therapy has proven to be effective in many patients with severe eosinophilic asthma. In this observational multicenter retrospective study, we report the results obtained with mepolizumab and benralizumab in severe asthmatics treated for 12 months in a real-life setting. In these patients, peripheral eosinophil levels, pulmonary function trends, exacerbation rates, systemic corticosteroid use, and symptom control were evaluated during the observation period, to understand which patients met all the criteria in order to be considered in disease remission. The percentage of remittent patients was 30.12% in the mepolizumab-treated subgroup, while in the benralizumab-treated subgroup, patients in complete disease remission were 40%, after 12 months. The results of this study confirm the efficacy of anti-IL-5 biologic drugs in the treatment of severe eosinophilic asthma in a real-life setting.

## 1. Introduction

Asthma is a chronic respiratory disease characterized by variable symptoms of wheezing, shortness of breath, chest tightness, cough, and variable expiratory airflow limitation, and is usually associated with airway hyperresponsiveness to direct or indirect stimuli and chronic inflammation of the airways [1]. This widespread airway disorder affects more than 300 million asthmatics worldwide and its prevalence is increasing [2]. Although most patients achieve good symptom control using drugs that have been available for decades, such as inhaled corticosteroids and bronchodilators, there is a percentage of patients ranging from 5 to 10% who, despite optimized treatments, do not reach a satisfactory clinical control, requiring the use of additional resources both in terms of direct and indirect costs. 

In recent years, the deepening of knowledge on the etiopathogenetic mechanisms underlying the different phenotypes and endotypes of asthma has led to the possibility of developing extremely refined pharmacological treatments, capable of intercepting specific molecular inflammatory pathways, with the potential aim of modifying the natural history of the disease. Although most of these drugs are currently available for the treatment of type-2 asthma endotype, drugs that target the so-called alarmins have very recently been developed, which act upstream of the type 2 inflammatory cascade [3].

The diffusion and use of biological therapies has allowed clinicians to consider asthma no longer simply a treatable disease, but potentially curable. For this reason, similarly to what has already happened for treatments relating to pathologies, such as rheumatoid arthritis, biological drugs used in severe asthma have been indicated as “disease modifiers” [4].

The ability of current drugs to determine substantial changes in the daily lives of patients who use them has allowed clinicians to identify the best outcomes capable of quantifying the degree of response to these treatments. Among the response parameters, in previous studies, some elements relating to lung function, the concentration of specific biomarkers, the use of concomitant oral corticosteroid therapy and the clinical response measured through validated questionnaires have been included [5,6,7].

In recent years, numerous real-life studies, mainly focusing on anti-IL5 and anti-IL5r pharmacotherapeutic strategies, have also been conducted with the aim of identifying categories of patients with severe asthma who could be considered super responders, and what the clinical, functional and biological characteristics of these patients were [8,9,10,11,12].

In this observational multicenter retrospective study, we report the results obtained with mepolizumab and benralizumab in severe asthmatics treated for at least 12 months in a real-life setting.

## 2. Results

Our retrospective observational study analyzed 113 severe asthmatics, of which 83 were treated with mepolizumab and 30 were treated with benralizumab, followed up for at least one year from the start of treatment. Some clinical and biological information was not available for all patients at the time of collection due to the retrospective nature of the study. The baseline characteristics of our patients are reported in Table 1. Our study population was predominantly female (36.28% were male patients), with a mean age of 57.5 years and a mean BMI of 27.2 kg/cm^2^. Most patients had no smoking history (65, 57.52%). The most reported comorbidities were chronic rhinosinusitis with nasal polyposis (42.47%) and gastroesophageal reflux disease (35.39%). All patients received high-dose inhaled corticosteroids (ICS) and a combination long-acting β2-agonist (LABA). All benralizumab-treated patients received OCS maintenance treatment with a mean dose of 14.6 mg/day prednisone-equivalent, whereas 63.8% of mepolizumab-treated patients received OCS maintenance treatment with an average OCS dose of 12.4 mg/day prednisone-equivalent. All patients had a history of poorly controlled asthma, as suggested by an ACT score < 20 (mean value 14.3 for mepolizumab-treated patients and 13.9 for benralizumab-treated patients), with frequent exacerbations despite maximal therapy (pooled mean 4.99 ± 2.69 in the previous year). In addition, all patients had airflow limitation, most often of moderate degree (predicted mean FEV1 63.62% for mepolizumab-treated patients and 54.57% for benralizumab-treated patients). For blood eosinophils, the mean baseline eosinophil count was 584.3 for mepolizumab-treated patients and 867.7 for benralizumab-treated patients. Other causes of hypereosinophilia were excluded in six patients with blood eosinophil counts > 1500 cells/μL prior to initiation of mepolizumab or benralizumab treatment.

All the data relating to the progression of the ACT score, of pulmonary function parameters and of exacerbation rate of the study population are reported in Table 2. Given the extreme heterogeneity of the population in study, it was decided to divide the patients depending on the treatment they received.

### 2.1. Blood Eosinophil Counts

Blood eosinophil counts significantly decreased throughout the observation period in mepolizumab-treated patients, from 584.3 ± 333.6 at the baseline to 101.5 ± 82.72 after 6 months of treatment (*p* < 0.0001) and 91.85 ± 112.9 after 12 months (*p* < 0.0001) (Figure 1A).

For patients treated with benralizumab, the peripheral blood eosinophil count decreased from the mean baseline value of 867.7 ± 537.9, passing to zero 6 months after the start of treatment (*p* < 0.0001) and remaining at zero throughout the duration of the study (*p* < 0.0001) (Figure 1B).

### 2.2. Pulmonary Function

Treatment with biological drugs resulted in a significant improvement of all pulmonary function parameters. 

Regarding the patient population treated with mepolizumab, there was a progressive improvement in all the lung function parameters taken into consideration. In these patients, FEV1 increased from 63.62 ± 17.13% at the baseline to 71.99 ± 16.57% after 6 months of treatment and finally to 73.91 ± 16.95% after 1 year of treatment, in both cases improving statistically significantly compared to the baseline.

Similarly, in patients treated with benralizumab there was a significant improvement in FEV1 during the observation period, with values going from 54.57 ± 14.7% at the baseline, to 76.7 ± 16.9% after 6 months of treatment, to 77.8 ± 16.5% after 1 year.

Likewise, all other pulmonary function parameters experienced statistically significant increases from baseline values in both study cohorts. The respective values are shown in Table 2 and in Figure 2.

### 2.3. Exacerbations and Use of Systemic Corticosteroids

Mepolizumab and benralizumab significantly reduced asthma exacerbations in their respective cohorts.

In the mepolizumab-treated subset of patients, the annualized exacerbation rate decreased from 4.27 ± 2.57 to 0.71 ± 1.05 and 0.39 ± 0.80 after 6 and 12 months of treatment, respectively (Table 2, Figure 3A).

Similarly, in the subgroup of benralizumab-treated patients, the annualized exacerbation rate decreased from 6.67 ± 2.2 to 0.13 ± 0.35 after 6 months, to 0.2 ± 0.48 after 12 months of treatment (Table 2, Figure 3C).

Regarding the use of oral corticosteroids, after 6 months of treatment, users of OCS had decreased from 63.8 to 25.3%, down to 18.07% after 12 months, in the cohort of patients treated with mepolizumab. This result was equally significant in the cohort of patients treated with benralizumab, in which it went from 100% at the baseline, to 10% after 6 months, up to 16.6% after 12 months of therapy.

For patients who continued OCS therapy, the mean dose used increased from 12.4 mg ± 11.9 to 1.36 mg ± 3.55 after 12 months in the mepolizumab subgroup (Table 2, Figure 3B), and decreased from 14.6 mg ± 8.75 to 1.1 mg ± 2.8 during the same observation period in the benralizumab subgroup (Table 2, Figure 3D).

### 2.4. Asthma Control Test

Treatment with biologics improved asthma control. After 12 months of therapy, asthma was well controlled in 59.03% of patients (ACT ≥ 20) treated with mepolizumab, with a mean increase in ACT score of 6.3 points from the baseline (Table 2, Figure 4A).

This result was even more striking in the subset of patients treated with benralizumab, in which patients with well-controlled asthma rose from 0% at the baseline to 83.3% after 12 months of treatment, with an average increase of 8.5 points in the mean ACT score from the baseline (Table 2, Figure 4B).

### 2.5. Asthma Clinical Remissions

After 12 months of treatment with biological drugs, the therapy was maintained in all cases. At the first 6-month follow-up, 14 of 83 patients (16.86%) in the clinical mepolizumab subgroup, and 11 of 30 patients (36.67%) in the benralizumab subgroup met all requirements to be considered in remission. The number of patients in remission varied further in the mepolizumab subgroup after 12 months, reaching a total of 25 (30.12%), while in the benralizumab subgroup, patients in complete disease remission were 12 (40%).

In general, the effect of the biological therapy had triggered changes in at least one of the parameters taken into consideration in the totality of the sample, and most of the beneficial effects were observed in terms of reduction in the use of oral corticosteroids and in the rate of exacerbations (Figure 5).

## 3. Discussion

In this real-life study, all patients treated with adjunctive anti-IL5 or anti-IL5r biologic therapy for severe eosinophilic asthma had a favorable response after one year of observation, with clinical and functional benefits. At the end of the observation period, complete remission, understood as the achievement of all four endpoints previously defined as absence of exacerbations, suspension from the use of OCS, ACT ≥ 20, and FEV1 ≥ 80% of the predicted, was reached in 30.12% of patients in the mepolizumab subgroup and 40% of patients in the benralizumab subgroup. Differently from other similar studies [10,13], no therapy switches were recorded during the observation period. 

In particular, there are two relevant aspects found following treatment with anti-IL5 and anti-IL5Rα. 

The first concerns the reduction of the average dosage of OCS among patients who used it continuously, as well as its suspension. Indeed, in both groups there was a marked reduction in the mean daily dosage of OCS, going from 12.4 mg to 1.36 mg and from 14.6 mg to 1.1 mg after one year of observation, respectively, in the subgroups treated with mepolizumab and benralizumab. In addition, the proportion of patients using corticosteroids decreased in one year from 63.8 to 18.07% among mepolizumab-treated patients and from 100 to 16.6% among benralizumab-treated patients. 

The second aspect concerns the marked reduction in the exacerbation rate over the observation period. It is interesting to note that these improvements were already evident after 6 months of therapy, and progressed after 12 months of treatment. In particular, an almost complete zeroing of the exacerbation rate was observed in both subgroups, going from 4.27 to 0.39 in patients treated with mepolizumab and from 6.67 to 0.2 in patients treated with benralizumab, after 1 year of observation.

These findings appear to be in line with what has already been highlighted in the recent literature [12,14,15,16,17,18].

Considering the aspects relating to the clinical outcomes and lung function individually, it is interesting to note that, beyond the results, the patients treated in both subgroups perceived an improvement, since none of them suspended or modified the therapy set at the beginning of the observation period. However, in terms of clinical improvement, the data relating to the percentage of patients with controlled asthma (defined by an ACT score ≥ 20) are significant, increasing from 15.66 to 59.03% for patients treated with mepolizumab and even from 0 to 83.3 % for benralizumab-treated patients, after one year of observation. The improvements in lung function were measured on the basis of parameters collected through forced post-bronchodilator maneuver; in particular an improvement in FEV1 was highlighted from 63.62 to 73.91% in the mepolizumab-treated subgroup and from 54.57 to 77.8% in the subgroup treated with benralizumab, after 1 year of observation. Additionally, FEF25-75, commonly used in clinical practice as an indirect index of the caliber of the small airways, obtained consistent improvements, going from 32.65 to 45.3% and from 33.47 to 56.9%, respectively, in the subgroups treated with mepolizumab and benralizumab after 1 year of therapy.

The findings obtained from our retrospective study appear in line with previous data present in the literature [8,12,19,20,21,22], although the methods used to evaluate clinical control differ according to the various recent studies, and although there is no unanimous consensus on the utility of the FEF25-75 as a reliable functional index related to the distal airways.

Regarding the significance of clinical remission, although several studies have already incorporated in the definition of “remission” the degree of disease control, lung function, use of OCS and exacerbation rate as necessary parameters to be evaluated in combination, differences were observed both in the cut-offs considered and in the responses evaluated following variable treatment periods [6,7,9,10,17,23,24,25,26,27]. Compared to these studies with a similar design, the percentage of remissions observed by us in the two subgroups is higher, reaching 30.12% after 1 year in patients treated with mepolizumab and 40% in patients treated with benralizumab, even considering the limits of the retrospective collection and the different methods of archiving data between the various centers that were involved in this study. 

The retrospective nature of the study showed that patients in the benralizumab-treated subgroup had more severe conditions at the baseline (as shown in Table 2 and Figure 5), with an average more compromised lung function, less disease control, the presence of higher peripheral blood eosinophilia, and higher exacerbation rates. Furthermore, although the mean OCS dosage did not differ significantly between the two subgroups under observation, the percentage of patients using OCS was higher in the benralizumab subgroup (100% versus 63.8%). These baseline characteristics reflect a trend by physicians to prescribe benralizumab in daily clinical practice to patients who were, on average, more functionally impaired and with higher biomarker levels, and this may bias the identification of patient populations to be observed retrospectively.

However, the aim of our analysis was to evaluate the effects of pharmacotherapeutic strategies currently approved in Italy to counteract the effects of IL-5 (mepolizumab and benralizumab) on a heterogeneous patient population and to investigate the outcomes over time, rather than to compare the subgroups with each other.

The heterogeneity of the patients of this study reflects the real-life setting and can be considered a strength, since it constitutes the main element of differentiation from randomized controlled trials. Furthermore, the size of the sample allows more reliable observations to be made, even considering the 12-month observation duration. However, it will be useful to implement new prospective studies in which it will be possible to collect homogeneous and systematic data over longer observation periods. 

Finally, a further element of novelty in our work is constituted by the criteria considered for clinical remission, differing from similar previous works especially in the aspect relating to the normalization of lung function (measured through FEV1, with values that were greater than or equal to the 80% of the predicted) and in the control of disease symptoms (measured through the scores obtained from the ACT questionnaire, most used in daily clinical practice).

The main limitations of this study, on the other hand, lie mainly in its retrospective design. In particular, the lack of uniformity in data collection and the possibility of having missing data are certainly recurring problems in observational studies involving multiple centers; however, in our case, we tried to keep the data collected during our visits as uniform as possible, since our centers usually collaborate in the analysis of patient populations affected by severe asthma and treated with biological drugs.

Once again, we want to specify that this study was not intended to compare the efficacy or superiority of one biologic drug over another, also because a randomized head-to-head comparison would be more appropriate for this purpose.

Some open questions remain, above all relating to the possibility of maintaining the state of remission, which, moreover, is reached early in our study population, following the suspension of the treatment.

The interruption of therapy with biologics is currently an aspect that still needs to be explored. Although there is evidence in the literature documenting that after discontinuation, asthma control is reduced [28,29,30], potentially nullifying the condition of clinical remission, it is not yet clear whether there are clusters of patients who could successfully suspend biological therapy without worsening, and above all, it is not yet clear on the basis of which biological characteristics it is possible to identify this cluster. Furthermore, the comorbidity profile of these patients must be considered, since the control of the concomitant pathologies contributes to the control of the asthmatic condition.

There are studies that have tried to define useful algorithms in an attempt to interrupt therapy with biological drugs; a recent study proposed no asthma symptoms (ACQ score < 1.5 or ACT score > 19), no asthma exacerbations, no use of oral corticosteroids, normalized spirometry (FEV1 > 80%), type 2 inflammation (blood eosinophil count < 300 mL and FeNO < 50 ppb), and control of comorbidities as possible discontinuation criteria [26]. 

Another aspect of great interest is to understand how long after clinical-functional remission of severe asthma it should be possible to attempt to suspend biological therapy. At the moment, there are no such elements to be able to answer this question. Further in-depth studies will be needed to establish whether remission can be sustained after discontinuation of the biologic drug.

## 4. Materials and Methods

For this retrospective study, data relating to 113 patients suffering from severe asthma and treated with mepolizumab or benralizumab were considered, in particular, previous data from 83 patients who had completed at least 12 months of continuous therapy with mepolizumab and 30 with benralizumab were evaluated. In addition to eligibility for mepolizumab and benralizumab, other criteria for study participation were the absence of other prior biologic therapies for severe asthma, the exclusion of concurrent respiratory disease, and other conditions requiring maintenance steroid therapy. The observation period included the last 3 years, in order to include as many patients as possible who had completed at least one year of treatment. The data of the study subjects were collected in 3 centers in southern Italy (Catanzaro, Naples, Salerno). During the data collection phase, 16 patients treated with mepolizumab and 7 treated with benralizumab were excluded from the study, due to an insufficient amount of data collected. Baseline characteristics of the study population are reported in Table 1.

Data were collected including the maximum number of records, taking into account the completeness of retrospectively provided information without performing any pre-study power analysis; however, the total number of patients included is similar to, and sometimes exceeds, the number of candidates included in similar real-life studies in the literature.

The diagnosis of severe asthma was made according to the European Respiratory Society (ERS)/American Thoracic Society (ATS) guidelines and eligibility for mepolizumab or benralizumab treatment was assessed according to the Italian Drug Agency (AIFA)’s prescription criteria. 

None of patients included in the study had previously received biologics for the treatment of SEA. Mepolizumab was administered subcutaneously at a dosage of 100 mg every 28 days, while Benralizumab was administered subcutaneously at a dosage of 30 mg every 4 weeks for the first three doses and every 8 weeks thereafter. 

Follow-up assessments occurred at 6 and 12 months of treatment and included clinical and biological data collection, and pulmonary function tests. Asthma control test (ACT) score, number of exacerbations and need of maintenance systemic corticosteroid were recorded as clinical outcomes. 

Exacerbations considered were those with worsening of symptoms, requiring oral corticosteroids or an increase from a stable maintenance dose, for at least three days. 

Blood eosinophil count was recorded as biological outcomes, and FEV1%, FVC%, and FEV1/FVC% were reported as parameters of pulmonary function (Table 2).

The considered definition of clinical asthma remission follows the one proposed by Menzies-Gow et al. in 2022 [11], which evaluated the use of OCS, patient-reported asthma control (ACQ-6) and lung function (FEV1) as parameters exacerbations [6,31]. However, in consideration of the retrospective data available in the study population, it was decided to modify the criterion relating to asthma control, taking into consideration the ACT score instead of the ACQ-6 score. Furthermore, in consideration of the wide variability of the FEV1 values measured at the baseline, we wanted to consider FEV1 values that were greater than 80% of the predicted, differently from what had been proposed in the original study. Responses to each of these criteria were defined as follows: zero exacerbations, zero use of OCS, ACT score > 20, and pre-bronchodilator FEV1 > 80% predicted.

All the patients gave informed consent for the use of their personal data. This observational study was undertaken in accordance with the Helsinki Declaration and was approved by the Local Ethical Committee. Statistical analysis was performed using Prism Version 9.2.1 (Graphpad Software Inc., San Diego, CA, USA). The data were reported as mean and standard deviation (SD) for normally distributed data and median and interquartile range (IQR) for skewed distributed data. The categorical variables were considered as the number of cases and percentages. When appropriate, variable comparations were performed using the Friedman test or a mixed-effects model. Probability values of <0.05 were considered to be statistically significant.

## 5. Conclusions

The results of this study confirm the efficacy of biologics in the treatment of severe eosinophilic asthma in a real-life setting.

Add-on therapy with benralizumab and mepolizumab resulted in significant functional and clinical improvements, such as increased airway caliber, symptom control, decreased exacerbation rate, and OCS reduction, in severe eosinophilic asthmatic patients. It is of particular relevance that the achievement of all these objectives was obtained in a significant percentage of the patients observed, allowing them to be considered in remission of the disease. Fundamental critical issues for future research are the identification of predictors of optimal response to biological treatment.

## Figures and Tables

**Figure 1 ijms-24-02455-f001:**
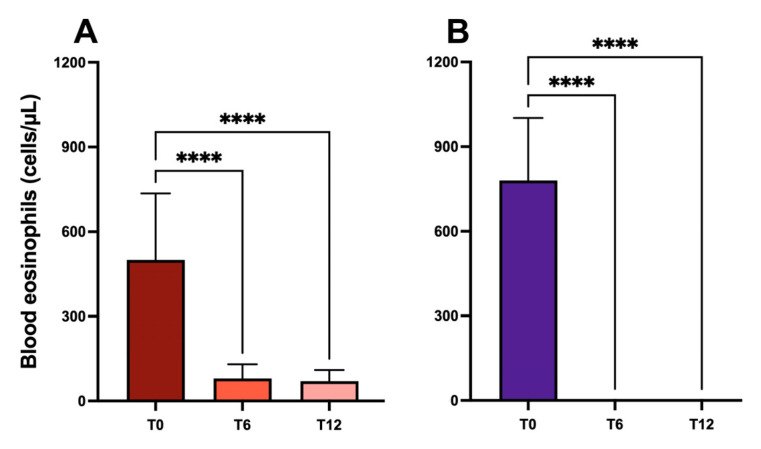
Effects of mepolizumab (**panel A**) and benralizumab (**panel B**) on peripheral blood eosinophil count (****: *p* < 0.0001).

**Figure 2 ijms-24-02455-f002:**
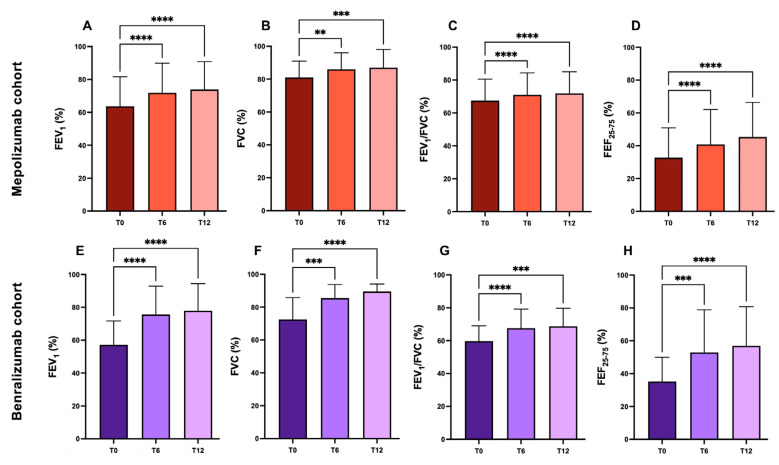
Improvements in lung function parameters after treatment with biologics. Panels (**A**–**D**) refer to mepolizumab-treated patients; panels (**E**–**H**) refer to benralizumab-treated patients (**: *p* < 0.05; ***: *p* < 0.0005; ****: *p* < 0.0001).

**Figure 3 ijms-24-02455-f003:**
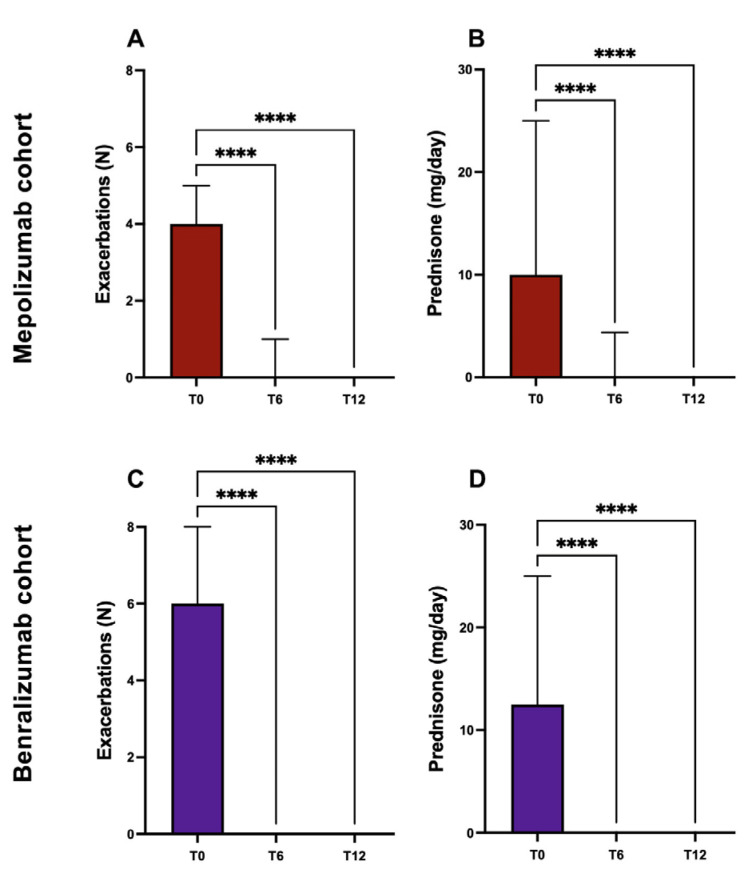
Effects of mepolizumab (**panels A,B**) and benralizumab (**panels C,D**) on exacerbations and OCS mean daily dose (****: *p* < 0.0001).

**Figure 4 ijms-24-02455-f004:**
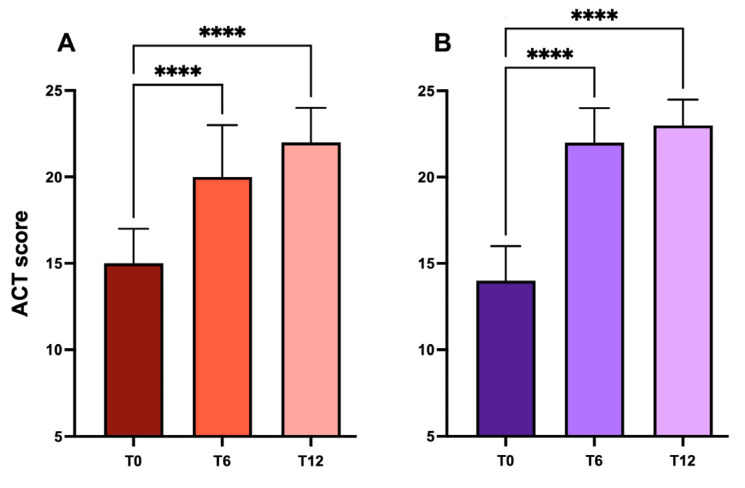
Effects of mepolizumab (**panel A**) and benralizumab (**panel B**) on ACT score over time (****: *p* < 0.0001).

**Figure 5 ijms-24-02455-f005:**
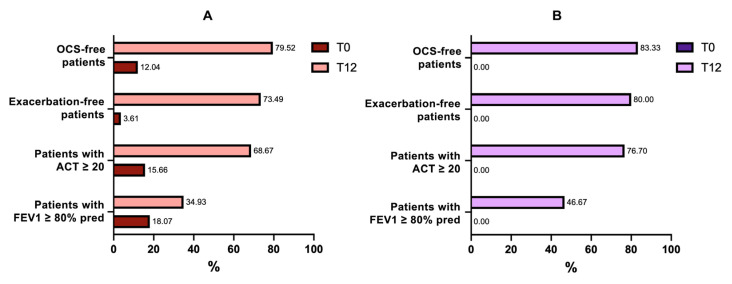
Percentage of mepolizumab-treated (**panel A**) and benralizumab-treated (**panel B**) patients meeting individual criteria for clinical remission in asthma.

**Table 1 ijms-24-02455-t001:** Baseline characteristics of the study population (m: mepolizumab-treated; b: benralizumab-treated).

Patients, n	113 (83 m; 30 b)
Age (mean ± SD)	57.5 ± 8.9
Gender (male)	41 M (36.28%)
Smokers/former smokers/ non-smokers	16/32/65
Body mass index, kg/m^2^ mean ± SD	27.2 ± 3.4
Asthma duration median (IQR)	21.5 (18.75)
Age at asthma onset Mean ± SD	35.9 ± 10.9
OCS-dependent patients	83 (75.45%)
Comorbidities	
Obesity (BMI ≥ 30 kg/m^2^)	26 (23%)
Chronic rhinosinusitis with nasal polyposis	48 (42.47%)
Gastroesophageal reflux disease	40 (35.39%)

**Table 2 ijms-24-02455-t002:** Clinical, biological, and functional data of the study population.

	Treated with Mepolizumab (n = 83)			Treated with Benralizumab (n = 30)		
	T0	T6	T12	T0	T6	T12
ACT score	14.3 ± 4.8	19.4 ± 4.2	20.6 ± 4.5	13.9 ± 2.8	21.9 ± 2.9	22.4 ± 2.6
ACT ≥ 20	13 (15.66%)	39 (46.98%)	49 (59.03%)	0 (0%)	23 (76.6%)	25 (83.3%)
Blood eosinophils (cells/μL) mean ± SD	584.3 ± 333.6	101.5 ± 82.72	91.85 ± 112.9	867.7 ± 537.9	0	0
Exacerbation history, previous year (n/y)	4.27 ± 2.57	0.71 ± 1.05	0.39 ± 0.80	6.67 ± 2.2	0.13 ± 0.35	0.2 ± 0.48
OCS users, n	53 (63.8%)	21 (25.3%)	15 (18.07%)	30 (100%)	3 (10%)	5 (16.6%)
OCS (prednisone-equivalent mg/die) mean ± SD	12.4 ± 11.9	3.47 ± 6.83	1.36 ± 3.55	14.6 ± 8.75	1.17 ± 2.9	1.1 ± 2.8
FEV1 %th mean ± SD	63.62 ± 17.13	71.99 ± 16.57	73.91 ± 16.95	54.57 ± 14.7	76.7 ± 16.9	77.8 ± 16.5
FVC %th mean ± SD	80.26 ± 15.43	85.84 ± 15.33	86.63 ± 15.5	72.78 ± 14.5	86.3 ± 12.3	86.7 ± 13.1
FEV1/FVC %th mean ± SD	66.53 ± 12.89	70.2 ± 13.2	71.87 ± 13.2	59.21 ± 9.72	68.16 ± 11.88	68.6 ± 11
FEF25-75 %th mean ± SD	32.65 ± 16.64	38.87 ± 17.16	45.3 ± 21	33.47 ± 15.1	53.5 ± 27.9	56.9 ± 23.9

## Data Availability

Data are available on request due to privacy restrictions.

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
