# Peer review of "Severe Asthma Remissions Induced by Biologics Targeting IL5/IL5r: Results from a Multicenter Real-Life Study"

_ijms, 2023, doi:10.3390/ijms24032455_

Round 1
Reviewer 1 Report
In this study, they obtained results from severe eosinophilic asthma after treating with mepolizumab and benralizumab for 12 months in a real-life setting. In these they analysed some parameters like, peripheral eosinophil levels, pulmonary function trends, exacerbation rates and systemic corticosteroid which are correlate with the eosinophilic asthma. From this author concluded that efficacy of anti-IL-5 biologic drugs is a promising approach for remission of the severe eosinophilic asthma. This work is more appreciable and more helpful for the physicians those who are dealing with eosinophilic asthma patients.
Author Response
We warmly thank the reviewer for his interest in reading our work and for the precious feedback he has provided. Like the reviewer, we also sincerely hope that these results will be useful to colleagues who treat severe asthma with biologics in daily clinical practice. Thank you again.
Reviewer 2 Report
The manuscript submitted for review is a multi-center, retrospective, observational report on the real-world outcomes of a population of patients with severe eosinophilic asthma treated with two different biologics. The purpose of the study is clearly presented and the data is interesting, especially to treating clinicians. There are minor improvements needed.
Reword sentence lines 42-45 for clarity and grammatical structure.
Line 62 – “at least 12 months”? Is the word “least” missing from this sentence?
Materials and Methods:
The authors state the the case data presented was selected retrospectively from "3 centers in Southern Italy in the last three years". I am unclear regarding a couple aspects of this data.
Was a power analysis performed prior to the study to determine number of patients needed to investigate outcomes of interest with desired power? If so, please include this information.
What was the criteria used to select these cases? Do these data represent all of the cases of severe eosinophilic asthma treated with these biologics in this 3 year period? Were cases chosen randomly, if so what was the total eligible case number? Please clearly describe any inclusion/exclusion criteria used for case selection as this can help the reader better compare this patient population to their own patients.
The manuscript text and statements at the end of the manuscript regarding Ethical review and patient consent are not consistent. Please resolve with the correct information
In the manuscript it states: All the patients gave informed consent for the use of their personal data. This observational study was undertaken in accordance with the Helsinki Declaration and was approved by the Local Ethical Committee.
The declarations at the end of the manuscript state:
Institutional Review Board Statement: The study was conducted by analyzing retrospective data 344 obtained from various hospitals located in southern Italy. Ethical review and approval are not applicable.
Informed Consent Statement: Not applicable.
Author Response
We thank the reviewer very much for the interest shown in reading our work and for the valuable suggestions he has provided, which we enthusiastically integrate into the manuscript, reporting our point-by-point answers below:
Q: Reword sentence lines 42-45 for clarity and grammatical structure.
A: We have improved the grammatical structure and sentence form.
Q: Line 62 – “at least 12 months”? Is the word “least” missing from this sentence?
A: We actually made a mistake, and so added the missing word.
Q: Was a power analysis performed prior to the study to determine number of patients needed to investigate outcomes of interest with desired power? If so, please include this information.
A: We thank you for the suggestion and we proceeded to elaborate on the matter in lines 80-84.
Q: What was the criteria used to select these cases? Do these data represent all of the cases of severe eosinophilic asthma treated with these biologics in this 3 year period? Were cases chosen randomly, if so what was the total eligible case number? Please clearly describe any inclusion/exclusion criteria used for case selection as this can help the reader better compare this patient population to their own patients.
A: Also in this case, we thank the auditor for the careful and punctual reporting, and we proceed to include the missing data on lines 70-79
Q: The manuscript text and statements at the end of the manuscript regarding Ethical review and patient consent are not consistent. Please resolve with the correct information
In the manuscript it states: All the patients gave informed consent for the use of their personal data. This observational study was undertaken in accordance with the Helsinki Declaration and was approved by the Local Ethical Committee.
The declarations at the end of the manuscript state:
Institutional Review Board Statement: The study was conducted by analyzing retrospective data 344 obtained from various hospitals located in southern Italy. Ethical review and approval are not applicable.
Informed Consent Statement: Not applicable.
A: The reviewer is absolutely right and the error came from using an older template. We have corrected the statement with the right information, at the end of the manuscript.